# Irbesartan May Ameliorate Ventricular Remodeling by Inhibiting CREB-Mediated Cardiac Aldosterone Synthesis in Rats with Myocardial Infarction

**DOI:** 10.3390/ijms26010198

**Published:** 2024-12-29

**Authors:** Jie Li, Guihua Lu, Haiwei Deng, Xiuren Gao, Yuansheng Zhai

**Affiliations:** 1Department of Cardiology, The First Affiliated Hospital, Sun Yat-Sen University, Guangzhou 510800, China; lijie268@mail.sysu.edu.cn (J.L.); luguihua3@mail.sysu.edu.cn (G.L.); denghw26@mail.sysu.edu.cn (H.D.); 2Key Laboratory on Assisted Circulation, Ministry of Health, Guangzhou 510800, China

**Keywords:** ventricular remodeling, myocardial infarction, aldosterone, angiotensin II, CYP11B2

## Abstract

Irbesartan improves ventricular remodeling (VR) following myocardial infarction (MI). This study investigates whether irbesartan attenuates VR by reducing aldosterone production in the heart and its underlying mechanisms. MI was induced in male Sprague–Dawley rats through coronary artery ligation. The MI rats were randomly assigned to two groups: one received a vehicle, and the other received 100 mg/kg/day of irbesartan for 5 weeks. Cardiac function and myocardial fibrosis were assessed using echocardiography and Masson’s trichrome staining, respectively. The impact of angiotensin II (Ang II) stimulation on cardiac microvascular endothelial cells (CMECs) from commercial sources was determined using ELISA, real-time PCR, and Western blotting. Irbesartan reduced left ventricular mass index, collagen composition, and aldosterone levels while enhancing cardiac function in MI rats. In vitro, Ang II time-dependently stimulated aldosterone secretion and CYP11B2 mRNA expression in CMECs (*p* < 0.05). Additionally, Ang II significantly upregulated p-CREB protein levels. However, these effects were abrogated by irbesartan and partially attenuated by CaMK inhibitor KN93 (*p* < 0.05). In conclusion, our study demonstrated that improvement in VR by irbesartan coincided with reduced CREB phosphorylation in CMECs and reduced aldosterone synthesis in the non-infarcted tissue. These effects may be mediated by blocking the AT1 receptor.

## 1. Introduction

Acute myocardial infarction (AMI) is a leading cause of heart failure (HF), with a mortality rate comparable to that of cancer. Consequently, it imposes an immense financial burden on healthcare systems globally [1,2]. Ventricular remodeling (VR) is the key pathophysiological mechanism of HF following myocardial infarction (MI) and an independent predictor for poor prognosis, indicating the role of ameliorating VR for patients with MI [3].

VR following MI involves cell death, inflammation, and myocardial interstitial fibrosis, ultimately leading to ventricular insufficiency and changes in left ventricular geometry, such as cardiac enlargement and ventricular aneurysm [4]. Activation of collagen-producing myofibroblasts following MI may cause progressive myocardial fibrosis [5]. This fibrotic process is closely regulated by transforming growth factor-β1 (TGF-β1)/Smad 2 pathway and aldosterone [6,7]. It has been reported that TGF-β1 interacts with aldosterone in myocardial fibrosis [8]. BrigHTN Clinical Trials have revealed that aldosterone synthase inhibitors reduced blood pressure in patients with treatment-resistant hypertension. These findings suggest that inhibiting the synthesis of aldosterone may represent a novel therapeutic strategy for the mitigation of myocardial fibrosis [9].

Aldosterone produced by both the adrenal glands and the heart causes post-MI myocardial fibrosis. According to previous studies, cardiac aldosterone may exert greater effects on VR [10]. Consistent with other studies, our previous research supported the association between aldosterone and post-MI myocardial fibrosis. Cardiac microvascular endothelial cells (CMECs) could be an important source of aldosterone in the heart. Our preliminary research has identified the existence of aldosterone synthase, the rate-limiting enzyme involved in aldosterone production, in CMECs [11]. CYP11B2, the gene encoding aldosterone synthase, is dose-dependently upregulated by arginine vasopressin (AVP), possibly mediated by the phosphorylation of calmodulin kinase (CaMK) and cyclic adenosine monophosphate (cAMP)-response element-binding (CREB) protein [11]. Bassett et al. reported that angiotensin II (Ang II), another component of the renin–angiotensin–aldosterone system (RAAS), induced CYP11B2 expression after binding to AT1 receptor in the adrenal glands [12]. However, whether CREB is regulated by Ang II remains unclear.

It is widely acknowledged that Ang II causes VR through both direct and indirect mechanisms. Specifically, Ang II directly binds to the angiotensin type 1 (AT1) receptor, leading to collagen deposition [13]. Additionally, Ang II promotes aldosterone synthesis both in the adrenal glands and the heart. Hence, Ang II indirectly affects the heart via the mineralocorticoid receptor [12,14]. It has been reported that the AT1 receptor mediates Ang II-induced CYP11B2 expression in the adrenal glands. CaMK inhibitor KN93 or CREB mutation partially reduces CYP11B2 expression, indicating the involvement of CaMK and CREB in the regulation of CYP11B2 expression [12]. Hence, we can speculate that CaMK and CREB contribute to aldosterone production in post-MI myocardial fibrosis. In addition, our previous study showed that KN93 partially inhibited AVP-induced CYP11B2 expression in CMECs. So, whether KN93, a CaMK blocker, could affect Ang II-induced CYP11B2 expression in CMECs will be explored in the present study.

AT1 receptor blockers (ARBs), generally used as antihypertensive drugs, exhibit biological effects beyond lowering blood pressure [15]. They not only improve VR but also have beneficial effects on survival in HF patients [16]. However, it remains unclear whether the effect of ARBs on VR is mediated by inhibiting CREB-induced synthesis of cardiac aldosterone.

The present study aims to determine whether Ang II mediates aldosterone production through the phosphorylation of CREB and whether this pathway is associated with the inhibition of VR by irbesartan, an AT1 receptor blocker, in an attempt to obtain a new target for the treatment of VR following MI.

## 2. Results

### 2.1. Effects of Irbesartan on Left Ventricular Mass Index

Thirty rats were randomly assigned into three groups, with ten rats in each group. All rats except for one in the MI group survived until the end of the 5-week study. This paper does not present the results of the dead rats.

As shown in Table 1, untreated MI rats showed a notable increase in both the left ventricular mass index (LVMI, left ventricular mass/body weight) and left ventricular mass, compared with the results obtained from the sham-operated rats (*p* < 0.05). After a 5-week treatment period, irbesartan significantly decreased the LVMI and left ventricular mass in the MI rats (*p* < 0.05). This observation suggested that irbesartan may exert beneficial effects on reducing VR in MI rats (Table 1).

### 2.2. Effects of Irbesartan on Cardiac Function

Table 2 presents a comprehensive summary of findings related to cardiac function indicators. Compared to the sham-operated rats, MI rats exhibited a marked reduction in the left ventricular ejection fraction (LVEF) (*p* < 0.05), indicating a decline in cardiac contractility. Additionally, enlargement of the left ventricles was evident through an increase in left ventricular end-systolic diameter (LVESD), left ventricular end-diastolic diameter (LVEDD), left ventricular end-systolic volume (LVESV), and left ventricular end-diastolic volume (LVEDV) (*p* < 0.05). However, the administration of irbesartan resulted in a significant elevation of LVEF and attenuation of the size of the left ventricle in MI rats (*p* < 0.05), thus suggesting that irbesartan effectively mitigated the deterioration of cardiac function (Table 2 and Appendix A).

### 2.3. Effects of Irbesartan on Cardiac Interstitial Fibrosis

VR following MI is typically characterized by the development of cardiac interstitial fibrosis. Therefore, we investigated whether collagen deposition occurred in the non-infarcted region of the left ventricle, which is adjacent to the infarcted area. As depicted in Figure 1A, the myocardium and collagen were stained in red and blue by Masson’s trichrome staining, respectively. In the sham-operated rats, cardiomyocytes were of uniform size and shape and neatly arranged. In contrast, cardiomyocytes in the MI rats exhibited hypertrophy and a disordered arrangement (Figure 1A). Furthermore, there was an increase in the amount of collagen deposition in the myocardial interstitium in MI rats, as quantified by increased collagen volume fraction (CVF) (*p* < 0.05). Irbesartan significantly reduced collagen deposition, reflected by a decrease in the CVF (*p* < 0.05). Hence, it can be suggested that irbesartan might reduce cardiac interstitial fibrosis (Figure 1).

### 2.4. Effects of Irbesartan on Aldosterone Level in MI Rats

Compared to the sham-operated rats, a significant elevation in aldosterone levels was observed within the non-infarcted tissues of the left ventricle in the MI rats (*p* < 0.05). Notably, this increase was reduced by irbesartan (*p* < 0.05) (Figure 2A). Moreover, a correlation analysis revealed a positive association between aldosterone concentration and CVF (*p* < 0.05), indicating the pivotal role of aldosterone in the pathogenesis of cardiac fibrosis. (Figure 2B).

### 2.5. Effects of Irbesartan on the mRNA Expression of TGF-β1 and Smad 2

Compared to the sham-operated rats, all the MI rats exhibited a higher mRNA expression of TGF-β1 and Smad 2 in the non-infarcted tissues of the left ventricle (*p* < 0.05) (Figure 3). Nevertheless, irbesartan significantly decreased the mRNA expression of TGF-β1 and Smad 2 in MI rats (*p* < 0.05) (Figure 3).

### 2.6. Effects of Irbesartan on the mRNA Expression of CYP11B2 in CMECs

In comparison with the vehicle, 10^−7^ mol/L or 10^−8^ mol/L of Ang II significantly increased CYP11B2 mRNA expression (*p* < 0.05), whereas 10^−5^ mol/L or 10^−6^ mol/L of Ang II showed similar results (*p* > 0.05). Intriguingly, the mRNA expression of CYP11B2 reached its peak levels when CMECs were exposed to 10^−7^ mol/L of Ang II (Figure 4A). Furthermore, a time-dependent upregulation of the CYP11B2 mRNA expression was observed when 10^−7^ mol/L of Ang II was administered to CMECs for 12, 24, and 48 h (*p* < 0.05), respectively (Figure 4B). In addition, the Ang II-upregulated mRNA expression of CYP11B2 was completely abrogated when CMECs were pretreated with irbesartan (10^−5^ mol/L) (*p* < 0.05). In contrast, only a partial inhibition was observed when CMECs were pretreated with KN93 (10^−6^ mol/L) (*p* < 0.05) (Figure 4C).

### 2.7. Effects of Irbesartan on the Secretion of Aldosterone in CMECs

As shown in Figure 5, the administration of 10^−7^ mol/L of Ang II resulted in a greater secretion of aldosterone than that achieved through the administration of 10^−8^ mol/L of Ang II in CMECs (*p* < 0.05). In contrast, the production of aldosterone remained unaffected when CMECs were exposed to either 10^−5^ or 10^−6^ mol/L of Ang II (*p* > 0.05). We noted that aldosterone secretion was time-dependently induced when CMECs were treated with 10^−7^ mol/L of Ang II for 12, 24, and 48 h (*p* < 0.05). Next, we pretreated CMECs with irbesartan (10^−5^ mol/L) or KN93 (10^−6^ mol/L) before adding Ang II (10^−7^ mol/L). Our findings revealed that Ang II-induced aldosterone secretion was completely blocked by irbesartan, whereas KN93 exerted only a partial inhibitory effect (*p* < 0.05) (Figure 5).

### 2.8. Effects of Irbesartan on the Protein Levels of CREB in CMECs

As illustrated in Figure 6, the Ang II (10^−7^ mol/L)-treated group exhibited significantly elevated protein levels of the phosphorylated CREB (p-CREB) protein than that demonstrated by the vehicle group (*p* < 0.05). Notably, the protein levels of p-CREB in the irbesartan-treated group were similar to that of the vehicle group (*p* > 0.05), indicating that irbesartan effectively inhibited the Ang II-induced upregulation of p-CREB levels. Conversely, the administration of KN93 (10^−6^ mol/L) only partially blocked the Ang II-associated elevation in p-CREB levels (*p* < 0.05) (Figure 6).

## 3. Discussion

We reported four key findings: (1) the elevated expression of the TGF-β1 and Smad 2 coincided with elevated tissue aldosterone concentrations during the fibrotic process in the non-infarcted region of post-infarcted heart; (2) irbesartan improved myocardial fibrosis-induced VR and cardiac function, and this coincided with reduced local production of aldosterone; (3) Ang II induced CYP11B2 expression and aldosterone synthesis in CMECs by promoting the phosphorylation of CREB; and (4) irbesartan reduced aldosterone production and CYP11B2 expression by inhibiting the phosphorylation of CREB after binding to the AT1 receptor. The elucidation of this novel mechanism, through which irbesartan delays myocardial fibrosis by suppressing aldosterone production, offers innovative insights into the prevention and treatment of VR following MI.

Post-MI VR primarily comprises two distinctive stages: the early remodeling phase and the late remodeling phase [17]. The early VR serves as a physiological compensatory mechanism, healing the heart from myocardial injury and necrosis. However, during the late remodeling phase, unchecked stimulation of neuroendocrine hormones can cause adverse VR, ultimately culminating in an enlarged heart cavity and compromised heart function. In the present study, we observed that the hearts of the MI rats exhibited significant alterations in ventricular geometry and function compared with those of the sham-operated rats. Specifically, post-MI rats demonstrated increased LVMI, enlarged LVESD, LVEDD, LVESV, and LVEDV, and reduced LVEF. These findings effectively validate the establishment of MI animal models. It is widely acknowledged that myocardial interstitial fibrosis is a pathological hallmark of late remodeling [18]. Consistent with previous studies, our study also showed a significant increase in collagen deposition within the myocardial interstitium of post-MI rats, which contributed to adverse VR and a subsequent decline in cardiac contractile function [19].

ARBs, including irbesartan, were originally developed to reduce blood pressure. However, a growing body of research has reported a diverse array of favorable outcomes exhibited by these drugs among non-hypertensive patients, encompassing a range of cardiovascular conditions such as AMI and cardiomyopathy [20,21]. The beneficial cardiac effects of irbesartan may be attributed to its ability to modulate myocardial fibrosis. Our study showed that irbesartan significantly decreased CVF, an index of myocardial fibrosis, as well as LVMI in post-infarcted rats. Conversely, the LVEF of MI rats was observed to increase following 5-week treatment with irbesartan, indicating improved cardiac contractility. Furthermore, irbesartan was also found to significantly reduce aldosterone concentration in the non-infarcted tissues of the left ventricle in MI rats. Recent studies have highlighted the capability of ARBs to improve VR in normotensive patients without lowering their blood pressure, which could be owing to the direct inhibition of tissue RAAS [22,23]. Fraccarollo et al. reported that aldosterone receptor antagonism provided additional benefit to irbesartan on cardiac remodeling after MI in rats [24]. Accumulating evidence supports the existence of an aldosterone synthesis system within the heart, with aldosterone synthase detected not only in cardiomyocytes and vascular smooth muscle cells but also in CMECs [11,25,26]. Such locally produced aldosterone likely exerts its biological effects through autocrine and paracrine mechanisms. Our study showed that, compared to the sham-operated rats, AMI rats exhibited elevated mRNA expressions of TGF-β1 and Smad 2, which were significantly attenuated by irbesartan. Given that the TGF-β1/Smad signaling pathway has been implicated in mediating the fibrogenic effects of aldosterone in the heart, we speculated that irbesartan effectively modulated the expression of TGF and Smad 2 by suppressing the production of aldosterone, ultimately attenuating the accumulation of collagen and enhancing the cardiac contractile function.

The regulation of CYP11B2 expression serves as a fundamental step in determining the capability of the adrenal gland to synthesize aldosterone [27]. Although Ang II and K^+^ are recognized as the primary physiological regulators of CYP11B2 expression in the adrenal gland, very few studies have investigated the molecular mechanism underlying the transcriptional control of CYP11B2 by Ang II in hearts. We observed that Ang II time-dependently enhanced the expression of CYP11B2 mRNA and the secretion of aldosterone in CMECs. Unexpectedly, however, we failed to detect a proportional elevation in the mRNA expression of CYP11B2 and aldosterone secretion in response to increasing concentrations of Ang II, which aligned with our previous study results [11]. However, the precise mechanism underlying this phenomenon has not yet been elucidated. It is postulated that the growth of CMECs depends on a specific concentration range of Ang II. If the concentration exceeds an optimal threshold, it might inhibit cell proliferation and even lead to apoptosis or cell death. Nevertheless, additional research is warranted to understand this phenomenon.

Next, we focused on CaMK to identify the cellular factors that mediated the Ang II-induced production of aldosterone in CMECs. The multifunctional CaMK, a widely distributed Ca^2+^-dependent kinase, has been shown to promote adverse VR triggered by cardiac overload or MI [28,29]. Yao et al. reported that the inhibition of CaMK in cardiomyocytes attenuates cardiac ischemia/reperfusion injury, thereby subsequently alleviating VR [30]. Furthermore, CaMK modulates Ang II-induced smooth muscle cell hypertrophy in cultured aortic smooth muscle cells [31]. In contrast, KN93, a suppressor of CaMK, effectively blocks the biological effects associated with Ang II. Our study results also revealed that Ang II binds to the AT1 receptor and elicits the upregulation of CYP11B2 mRNA and the production of aldosterone. However, the effects were partially mitigated by KN 93, indicating the involvement of CaMK in Ang II-associated aldosterone synthesis in CMECs.

The cAMP response element (CRE), located within the CYP11B2 promoter, has been demonstrated to modulate transcription through its interaction with the members of the CREB family [32]. An analysis of the sequence has revealed that CRE maintained a remarkable degree of conservation across diverse species [33]. It held significant importance in regulating the CYP11B2 gene activity in bovines, rodents, and humans [34,35,36]. The elimination of CRE from the CYP11B2 5′-flanking region in human H295R adrenocortical cells resulted in a reduction in transcription levels upon cAMP stimulation [12]. Previous studies have also suggested the involvement of Ang II in the phosphorylation-induced activation of CREB across various tissues, including myocardial tissue. In vascular smooth muscle cells and H295R cells, CaMK has emerged as a key regulator of target gene transcription by phosphorylating CREB. Our preliminary study also reported that the activation of CaMK led to the phosphorylation of CREB and subsequent upregulation of CYP11B2 [11]. In the current study, similarly, Ang II was observed to synchronously enhance the expression of CYP11B2 and p-CREB when compared to that in the vehicle control. Nevertheless, these effects were comprehensively abrogated by irbesartan and partially attenuated by KN93. Hence, we postulated that Ang II might activate the CaMK/p-CREB pathway through the AT1 receptor in CMECs, ultimately contributing to the upregulation of CYP11B2 mRNA. Additionally, it was hypothesized that irbesartan might reduce the production of aldosterone by inhibiting the aforementioned pathway.

There are some limitations in the present study. First, given that post-MI cardiac fibrosis is affected by aldosterone produced from both the adrenal gland and the heart itself, it is difficult for us to determine to what degree this locally produced aldosterone is responsible for the fibrotic process [37,38]. Similarly, irbesartan may alleviate cardiac fibrosis by reducing aldosterone synthesis in both the adrenal gland and the heart. Therefore, the langendorff-isolated perfused rat heart model may be warranted to address these issues. Finally, aldosterone may originate not only from CMECs but also from cardiomyocytes, which constitute the primary cellular components of the heart [25]. Therefore, it is necessary for us to explore the mechanism by which Ang II participates in the regulation of aldosterone synthesis in cardiomyocytes in the future.

## 4. Materials and Methods

### 4.1. Animals and Irbesartan Treatment

We procured 30 male Sprague–Dawley rats weighing 250–350 g from the Medical Experimental Animal Center of Guangdong (Guangzhou, China). These animals were maintained in constant-temperature environments with 12 h light/darkness cycles and were provided with standard chow and water ad libitum. All experimental procedures were carefully reviewed and approved by the Experimental Animal Ethics Committee of Sun Yat-sen University (Guangzhou, China, SYSU-IACUC-2018-000281).

Following left thoracotomy and pericardiotomy, an MI rat model was produced by ligating the left anterior descending coronary artery [11]. A similar surgical procedure was performed in sham-operated rats without coronary artery ligation. After administering a 3-day course of penicillin through injection, the MI rats were randomly assigned to two groups. One received 100 mg/kg/day of irbesartan (Sanofi Pharmaceutical Company, Paris, France) dissolved in distilled water for 5 weeks. The other group received the vehicle (distilled water) for the same period. Sham-operated rats served as the control group and received distilled water for 5 weeks as well. Feeding was performed by oral gavage once a day.

The cardiac function of the animals was accurately evaluated using echocardiography following the treatment. At the end of the study, the rats were anesthetized with an intraperitoneal injection of 10% chloralhydrate at a dosage of 3 mL/kg, and their hearts were harvested. The excised hearts were then dried, and the left ventricular mass and the body weight were precisely measured. Subsequently, the LVMI was calculated by dividing the left ventricular mass by the body weight. For further analysis, the left ventricle, along with the septum, was transversely cut into three distinct parts: the apex, middle ring, and base, which were immediately either put into 10% neutral formalin for Masson’s trichrome staining or frozen in liquid nitrogen.

### 4.2. Echocardiography

Before the rats were sacrificed, one skilled operator, who was blinded to the group allocation, performed echocardiography in order to assess their cardiac function. Rats were weighed and anesthetized with 4.0% isoflurane. After the operator fixed it with tape and applied an ultrasonic coupler to the chest, a Vevo2100 ultrasound machine for animals (VisualSonics, Inc., Toronto, ON, Canada), equipped with a 21 MHz sector array transducer, was used to conduct the echocardiography. The operator captured the M-mode tracings of the long-axis view of the left ventricle and collected the following parameters: LVESD, LVEDD, LVESV, LVEDV, and LVEF. The data were presented as the average of three cardiac cycles.

### 4.3. Masson’s Trichrome Staining Analysis of Collagen Composition

Masson’s trichrome staining was performed to evaluate cardiac interstitial fibrosis. The middle ring of the left ventricle was fixed in 10% neutral formalin, dehydrated, embedded in paraffin, and then cut into 4 μm thick sections, which were used for Masson’s trichrome staining. Next, the stained sections were viewed with 100× magnification. Five non-infarcted fields were randomly selected in each section and analyzed using the Image J 1.53 Analytical System (National Institutes of Health, Bethesda, MA, USA). The CVF was calculated by dividing the collagen area by the total area of the microscopic field. The analyses were performed by one investigator blinded to group allocation, ensuring unbiased and reliable results.

### 4.4. Cell Culture In Vitro

We acquired primary rat CMECs from iCell Bioscience, Inc. (Shanghai, China). These cells were subsequently incubated at 37 °C in a humidified 5% CO_2_ atmosphere with Dulbecco’s modified Eagle’s medium/F12 (DMEM/F12, Gibco, USA) in the presence of 10% fetal bovine serum (FBS, Gibco, Carlsbad, CA, USA), which was supplemented with 1% penicillin/streptomycin. The third to fourth passages of CMECs were used in the present experiment. Cells were seeded in six-well plates at 6 × 10^5^ cells per well. After 48 h incubation, the cells were starved for 24 h and then treated with Ang II, irbesartan (AT1 receptor antagonist, Sigma, St. Louis, MO, USA), KN93 (CaMK antagonist, Sigma, St. Louis, MO, USA), or a vehicle for at least 12 h. The CMECs, which were cultured with DMEM/F12 in the presence of 10% fetal bovine serum without adding any drugs, were used as vehicles.

### 4.5. Enzyme-Linked Immunosorbent Assay (ELISA)

ELISA was performed to determine the aldosterone concentrations in the non-infarcted myocardial tissues of the heart, as well as in the cell culture medium of CMECs. The myocardial tissues were first grounded with a glass homogenizer in a phosphate-buffered saline solution (0.01 M, pH 7.4) and then centrifuged at 3000 rpm for 30 min. After the CMECs were incubated for 48 h, the cells were starved for 24 h and then treated with Ang II, irbesartan, KN93, or a vehicle for 24 h. Next, we collected the supernatant of the tissues and the CMECs to detect aldosterone concentrations. The aldosterone concentrations were quantitatively determined using commercially available ELISA kits (BlueGene Biotech Co., Shanghai, China) based on the instructions provided by the manufacturer.

### 4.6. Real-Time Polymerase Chain Reaction (PCR) Analysis

Real-time PCR was performed to measure the mRNA expressions of CYP11B2, TGF-β1, and Smad 2. Total RNA was extracted from homogenized non-infarcted myocardium tissues or the cell culture medium using TRIzol Reagent (Sigma, St. Louis, MO, USA). Then, 1 μg of total RNA was reverse-transcribed into complementary DNA (cDNA) using a commercially available reverse-transcription kit (Toyobo Co., Osaka, Japan). The produced cDNA was employed as a template for subsequent real-time PCR. The amplification products were measured using SYBR Green fluorescence (Vazyme Biotech Co., Nanjing, China). The forward primer and the reverse primer for rat CYP11B2, TGF-β1, Smad 2, and actin were as follows: CYP11B2 forward, 5′-ACC ATG GAT GTC CAG CAA-3′; CYP11B2 reverse, 5′-GAG AGC TGC CGA GTC TGA-3′; TGF-β1 forward, 5′-TGG TGG ACC GCA ACA ACG-3′; TGF-β1 reverse, 5′-GGC ACT GCT TCC CGA ATG-3′; Smad2 forward, 5′-TAC ATC CCA GAA ACA CCA CCA-3′; Smad2 reverse, 5′-GCG ATT GAA CAC CAA AAT GC-3′; actin forward, 5′-CGT TGA CAT CCG TAA AGA CCT C-3′; actin reverse, 5′-TAG GAG CCA GGG CAG TAA TCT-3′. The mRNA expressions of CYP11B2, TGF-β1, and Smad 2 were normalized with those of actin—an endogenous control gene.

### 4.7. Western Blotting

The protein levels of p-CREB in CMECs were assessed by Western blotting, as described previously [11]. Initially, total proteins were extracted from CMECs using a cell lysis buffer (Cell Signaling Technology, Danvers, MA, USA). Subsequently, a BCA assay (Pierce, Rockford, IL, USA) was conducted to measure the protein concentration. Next, aliquots (30 μg) were separated on 12% SDS gels by SDS-PAGE and then transferred onto PVDF membranes (Millipore, Bedford, MA, USA). The membranes were next blocked with 5% nonfat milk at room temperature for 1 h and then incubated with primary antibodies overnight at 4 °C. The following diluted concentrations of the primary antibodies were employed: p-CREB (Cell Signaling Technology, Danvers, MA, USA), 1:500; GAPDH (Servicebio Technology Co., Wuhan, China), 1:10,000. The goat anti-rabbit horseradish peroxidase-conjugated secondary antibodies were diluted to 1:10,000 and incubated at room temperature for 1 h. The protein bands were visualized with an enhanced chemiluminescence kit (Cell Signaling Technology, Danvers, MA, USA) and analyzed using the Image J 1.53 Analytical System. The relative band densities of p-CREB in Western blots were normalized against GAPDH.

### 4.8. Statistical Analysis

Statistical analyses were conducted utilizing SPSS 26.0 software (IBM Corporation, Armonk, NY, USA). All data were expressed as the mean ± SD. A Student’s *t*-test or non-parametric test was used for comparisons between the two groups. For multiple-group comparisons, a one-way analysis of variance (ANOVA) followed by a post hoc test or the Kruskal–Wallis test followed by Bonferroni analysis was performed. Correlations between two groups were analyzed using Pearson’s chi-square test or Spearman’s rank correlation test. A value of *p* < 0.05 was considered to be statistically significant.

## 5. Conclusions

In summary, our study demonstrated that irbesartan attenuated MI-induced myocardial fibrosis and VR induced by MI, which coincided with decreasing the synthesis of aldosterone in the heart. The cellular mechanism underlying these effects of irbesartan may involve the inhibition of CREB phosphorylation through the blockade of the AT1 receptor. These findings reveal that the suppression of the production of cardiac aldosterone might be employed as a novel approach to ameliorate VR. Further studies are needed to reveal to what degree this elevated local aldosterone synthesis contributes to the fibrotic process.

## Figures and Tables

**Figure 1 ijms-26-00198-f001:**
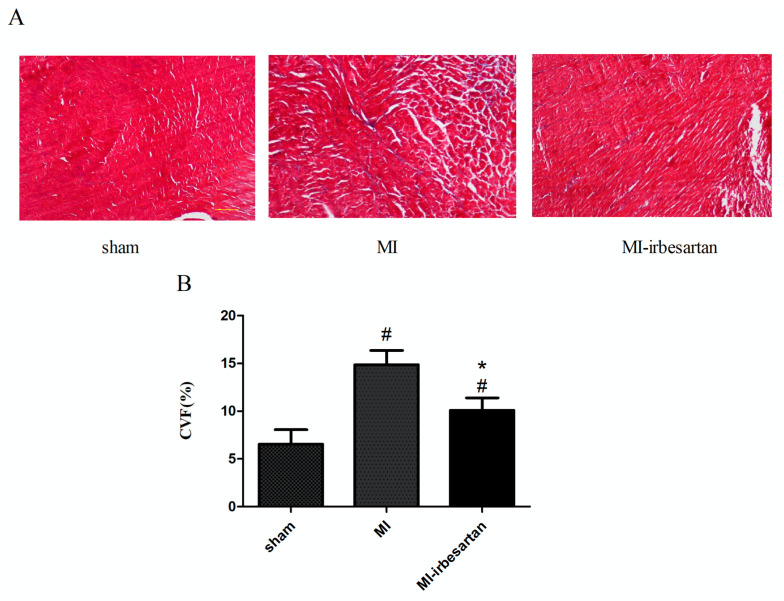
Effects of irbesartan on cardiac fibrosis. (**A**) Masson trichrome staining for collagen (magnification: ×100). Red indicates cardiomyocyte, and blue indicates collagen. sham, sham-operated rats; MI, myocardial infarction rats; MI–irbesartan, myocardial infarction rats fed with irbesartan. (**B**) Graphical representation of collagen volume fraction in the sections. CVF, collagen volume fraction; the results are presented as mean ± SD. # *p* < 0.05 vs. sham-operated rats; * *p* < 0.05 vs. myocardial infarction rats.

**Figure 2 ijms-26-00198-f002:**
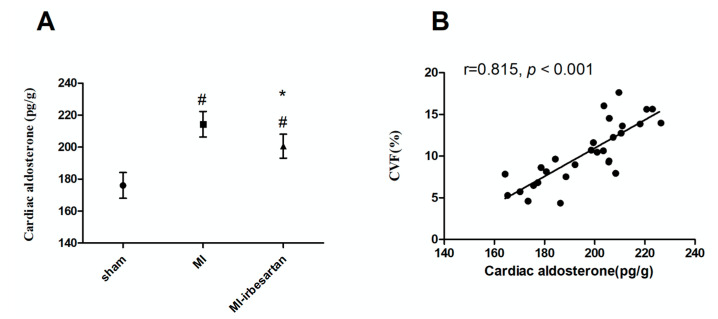
Effects of irbesartan on aldosterone levels in MI rats. The results are presented as mean ± SD. (**A**) Cardiac aldosterone levels. sham, sham-operated rats; MI, myocardial infarction rats; MI–irbesartan, myocardial infarction rats fed with irbesartan. # *p* < 0.05 vs. sham-operated rats; * *p* < 0.05 vs. myocardial infarction rats. (**B**) Correlation analysis between cardiac aldosterone and CVF. CVF, collagen volume fraction.

**Figure 3 ijms-26-00198-f003:**
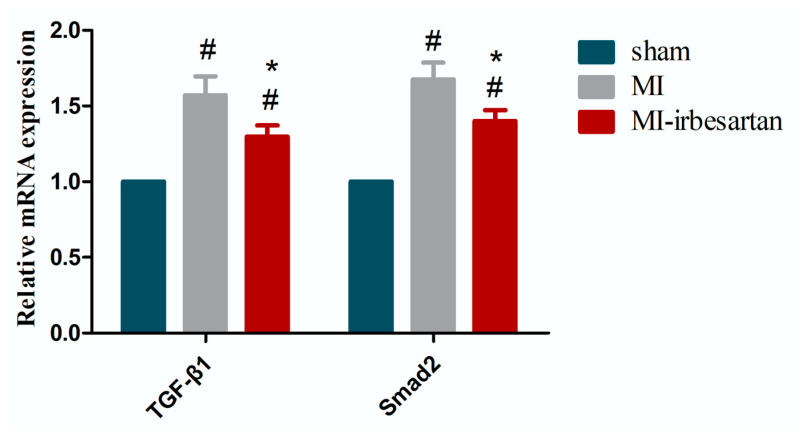
Effects of irbesartan on the mRNA expression of TGF-β1 and Smad 2. sham, sham-operated rats; MI, myocardial infarction rats; MI–irbesartan, myocardial infarction rats fed with irbesartan. The results are presented as mean ± SD. # *p* < 0.05 vs. sham-operated rats; * *p* < 0.05 vs. myocardial infarction rats.

**Figure 4 ijms-26-00198-f004:**
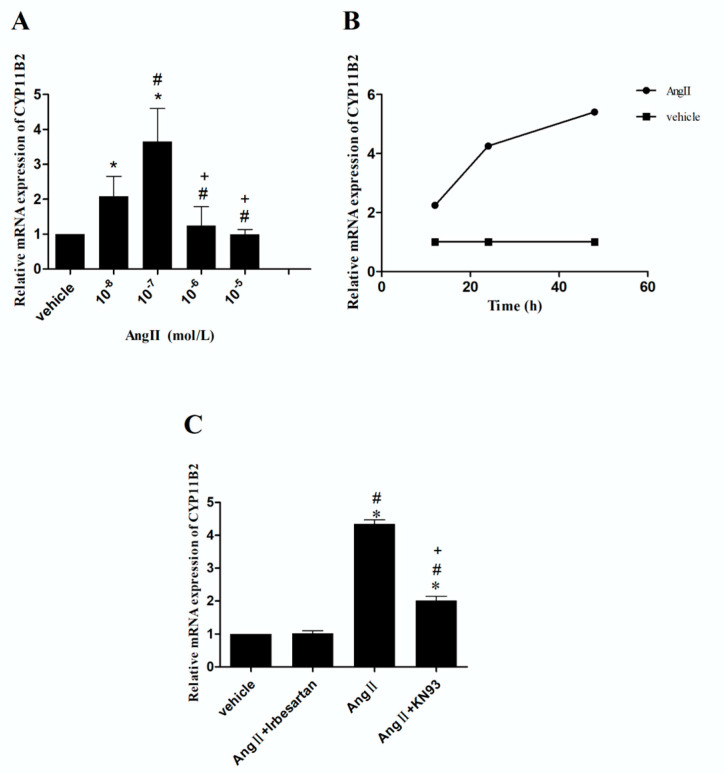
Effects of irbesartan on the CYP11B2 mRNA expression in CMECs. Five culture runs were performed for each group and three duplicates for every sample. (**A**) Effects of different concentrations of Ang II on CYP11B2 mRNA expression. * *p* < 0.05 vs. vehicle; # *p* < 0.05 vs. 10^−8^ mol/L Ang II; + *p* < 0.05 vs. 10^−7^ mol/L. (**B**) Ang II (10^−7^ mol/L) time-independently upregulated CYP11B2 mRNA expression. (**C**) Effects of different antagonists on Ang II-upregulated CYP11B2 mRNA expression. Ang II, 10^−7^ mol/L; irbesartan, 10^−5^ mol /L; KN93, 10^−6^ mol /L. * *p* < 0.05 vs. vehicle; # *p* < 0.05 vs. (Ang II + irbesartan); + *p* < 0.05 vs. Ang II.

**Figure 5 ijms-26-00198-f005:**
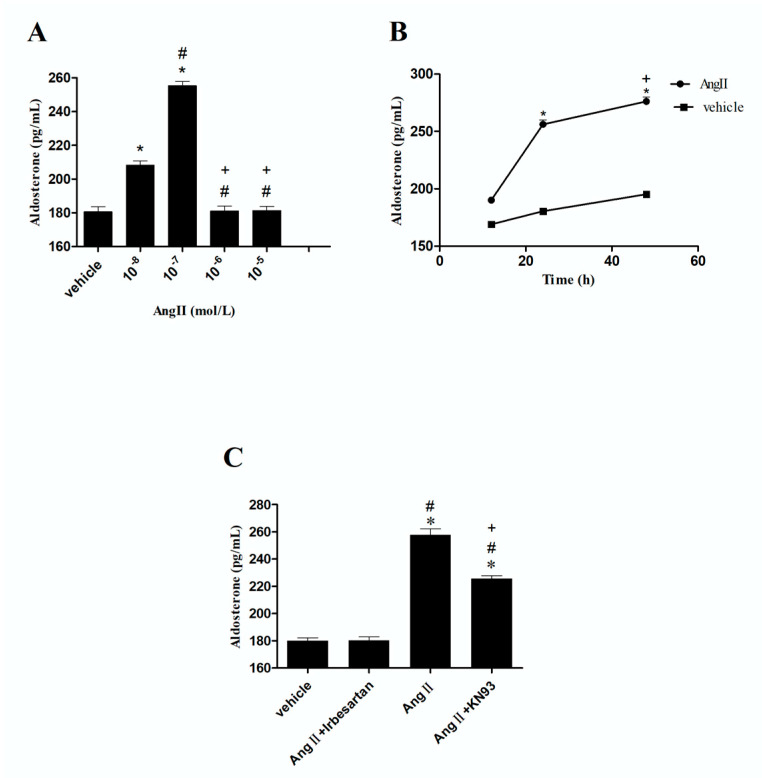
Effects of irbesartan on aldosterone secretion in CMECs. (**A**) Effects of different concentrations of Ang II on aldosterone secretion. * *p* < 0.05 vs. vehicle; # *p* < 0.05 vs. 10^−8^ mol/L Ang II; + *p* < 0.05 vs. 10^−7^ mol/L. (**B**) Ang II (10^−7^ mol/L) time-independently increased aldosterone secretion. (**C**) Effects of different antagonists on Ang II-induced aldosterone secretion. Ang II, 10^−7^ mol/L; irbesartan, 10^−5^ mol /L; KN93, 10^−6^ mol/L. * *p* < 0.05 vs. vehicle; # *p* < 0.05 vs. (Ang II + irbesartan); + *p* < 0.05 vs. Ang II.

**Figure 6 ijms-26-00198-f006:**
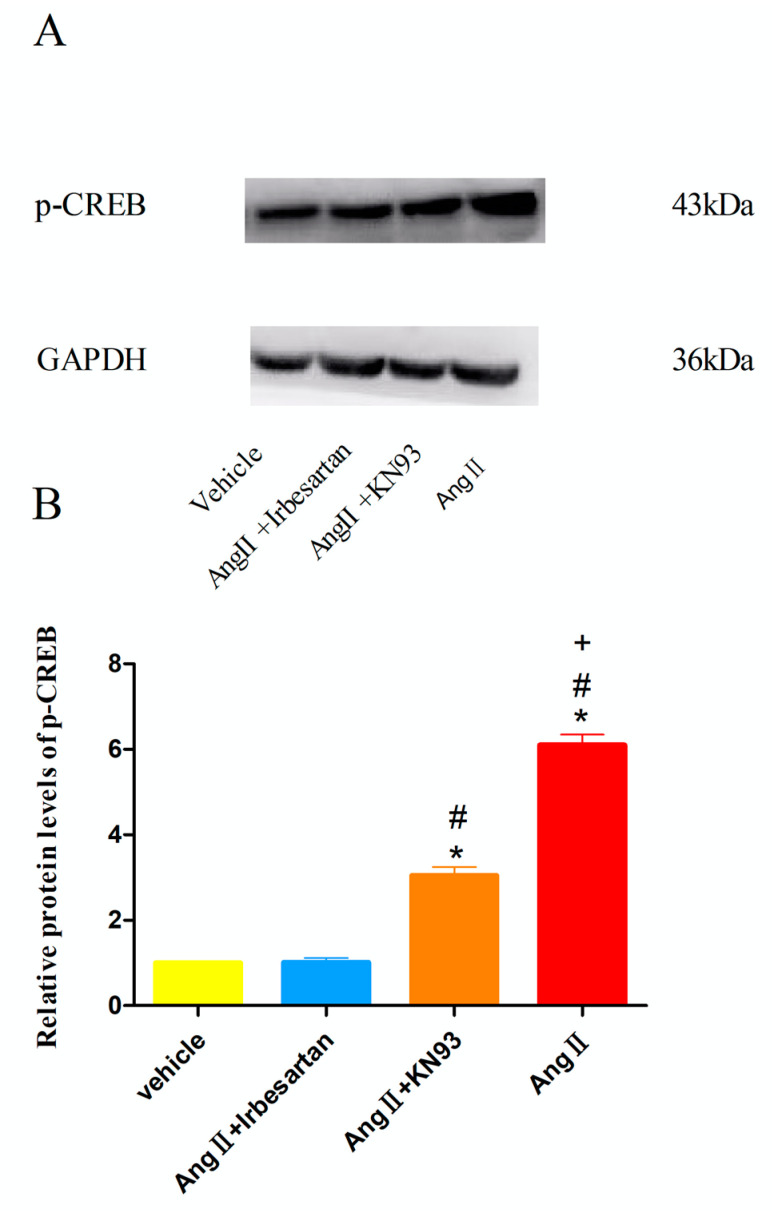
Effects of irbesartan on p-CREB protein levels in CMECs. p-CREB, phosphorylated CREB; Ang II, 10^−7^ mol/L; irbesartan, 10^−5^ mol /L; KN93, 10^−6^ mol /L. (**A**) Representative images of Western blots. (**B**) Graphical representation of protein levels of p-CREB. * *p* < 0.05 vs. vehicle; # *p* < 0.05 vs. (Ang II + irbesartan); + *p* < 0.05 vs. (Ang II + KN93).

**Table 1 ijms-26-00198-t001:** Left ventricular mass index.

Parameters	Sham	MI	MI–Irbesartan
	n = 10	n = 9	n = 10
BW (g)	417 ± 21	407 ± 22	410 ± 23
LVM (mg)	767 ± 21	837 ± 61 ^a^	791 ± 23 ^ab^
LVMI (mg/g)	1.84 ± 0.05	2.05 ± 0.08 ^a^	1.93 ± 0.06 ^ab^

BW, body weight; LVM, left ventricular mass; LVMI, left ventricular mass index; sham, sham-operated rats; MI, myocardial infarction rats; MI–irbesartan, myocardial infarction rats fed with irbesartan; data are presented as means ± S.D. ^a^ *p* < 0.05 vs. sham-operated rats; ^b^
*p* < 0.05 vs. myocardial infarction rats.

**Table 2 ijms-26-00198-t002:** Cardiac function.

Parameters	Sham	MI	MI–Irbesartan
	n = 10	n = 9	n = 10
LVESD (mm)	4.73 ± 1.32	7.45 ± 0.59 ^a^	6.37 ± 0.87 ^ab^
LVEDD (mm)	7.24 ± 1.22	9.18 ± 0.56 ^a^	8.36 ± 1.05 ^a^
LVESV (mm^3^)	114.7 ± 67.4	295.9 ± 52.8 ^a^	211.2 ± 64.7 ^ab^
LVEDV (mm^3^)	284.7 ± 101.7	469.4 ± 62.9 ^a^	387.0 ± 106.7 ^a^
LVEF (%)	62.7 ± 11.2	37.2 ± 4.7 ^a^	45.8 ± 2.1 ^ab^

LVESD, left ventricular end-systolic diameter; LVEDD, left ventricular end-diastolic diameter; LVESV, left ventricular end-systolic volume; LVEDV, left ventricular end-diastolic volume; LVEF, left ventricular ejection fraction; sham, sham-operated rats; MI, myocardial infarction rats; MI–irbesartan, myocardial infarction rats fed with irbesartan; data are presented as means ± S.D. ^a^ *p* < 0.05 vs. sham-operated rats; ^b^
*p* < 0.05 vs. myocardial infarction rats.

## Data Availability

Data are contained within the article and Appendix A.

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
