# Peer review of "Irbesartan May Ameliorate Ventricular Remodeling by Inhibiting CREB-Mediated Cardiac Aldosterone Synthesis in Rats with Myocardial Infarction"

_ijms, 2024, doi:10.3390/ijms26010198_

Round 1
Reviewer 1 Report
Comments and Suggestions for Authors
In the submitted paper the authors in a well organized and executed complex experimental in vivo and in vitro study demonstrated the elevated local peripheral aldosterone secretion during the fibrotization process in the infarcted rodent heart. However, one important flaw has remained in the logical buildup of the paper: we still do not know to what degree this locally produced aldosterone is responsible for the fibrotic process. This limitation should be included in form of a statement.
line 17 It has to be marked that these endothelial cells were commercial preparate, not from these infarcted animals
line 22 KN supposed effect of KN93 should be named
line 23 Improvement in ventricular remodeling by irbesartan coincided with reduced CREB phosphorylation, reduced aldosterone content and peripheral aldosterone synthesis in the infarcted tissue.
line 36 Among the consequences of myocardial infarct cardiac enlargement is only one among the several tens of serious symptoms (ventricular aneurysm? ventricular insufficiency?).
line 44 It has to be taken in consideration that in rodents the balance of glycol- and mineralocorticoids is different from that of humans: corticosterone released plays an important role.
line 39 Aldosterone is not a neuroendocrine hormone, correct the sentence to avoid that misleading implicit meaning.
lines 41-43, This pathway is not through a direct effect on myocardial cells.
line 53 The cAMP levels and thus CREB activation are controlled by many additional pathways in ventricular myocytes, AT1Rs being one of the less important pathways, the sentence has to be worded accordingly.
Results. The area affected by the coronary occlusion should be depicted.
line 110 The exact location of this noninfarcted region should be given (Close to the lesion? - potentially affected by reduced flow? Farher? . -affected by the heart failure.)
Fig 2. These tissue aldosterone concentrations are not too high, do not prove local synthesis. Elevated adrenal mineralocorticoid release can be expected in dilatative heart failure. (Aldosterone plasma concentrations in non-hypertensive patients are around 4 ng/dL (40 pg/ml) (Cannone V et al. Mayo Clin Proc 2018;93:980.).
line 140 Smad2 was not mentioned in the Introduction.
Fig 5. Aldosterone levels in the tissue culture (?) of coronary microvascular endothelial cells are hard to interpret in this form. Aldosterone levels in tissue culture of how many cells? In the superfusate? Produced amount in a given time?
Fig. 5. Application of the calcium dependent kinase type II blocker should be explained in the Introduction. line 183 Not phosphorylated CREBS is expressed, but CREBS. Existing, expressed CREBS molecules are activated by phosphorylation, the amount of which was measured. Correct the title of the subchapter.
line 189. What could be expected because of the existence of manyfold alternative activation pathways.
Discussion. It has to be rewritten, concluding that the expression of the TGFbeta pathway signalization peptide Smad2, phosphorylation of the CREB protein are elevated during the fibrotic process in the noninfarcted ventricular tissue of the infarcted rodent heart (known from earlier publications), and that it coincides with elevated expression of the aldosterone synthase enzyme and elevated tissue aldosterone concentrations. ATIIR1 inhibitor alleviates postischemic fibrosis and improves ventricular function (observations published earlier) and this coincides with reduced local production of this hormone.
line 340 Are these human or rodent cells?
Conclusion. Further studies are needed to reveal to what degree this elevated local aldosterone synthesis contributes to the fibrotic process.
The paper is linguistically flawless, correct scientific English, the Figures are constructed with high level of professionality.
Reviewer 2 Report
Comments and Suggestions for Authors
The authors explore the mechanisms underlying Ang-II driven remodeling in MI. It is fairly interesting paper overall. However, there are some major and minor problems.
Major:
1. Authors fail to mention an important paper on the similar topic (whether aldosterone blockade provide an additional benefit to Ang-II block): https://pubmed.ncbi.nlm.nih.gov/15949473/ That paper needs to be mentioned in Introduction and/or Discussion.
2. Presentation of the results: authors need to show individual data points not just averages ± SEM
3. MI example in Figure S1 is of poor image quality. I would suggest to use a larger images. Figure S1 can take a full page.
Minor
1. line 72: paper needs to mention at least once that irbesartan is Ang II receptor blocker
2. line 120: images looks oversaturated and the red color dominate everything. Is it possible to re-balance images, so the blue (fibrosis) is more visible?
3. line 124: CVF is somewhat misnomer since it is area-to-area and not volume-to-volume. I suggest using Fibrosis (%Area).
4. line 133: (A) is not representative but rather an average
5. line 160: please specify how many culture runs were performed and how many replicates per run.
6. line 183: term expression refers to mRNA levels. Here you probably meant protein levels. Please update related instances as well.
7. lines 394-395. The sentence "Further studies ...", in my opinion, is superfluous.
Reviewer 3 Report
Comments and Suggestions for Authors
In this study, Li and colleagues investigated the effects of irbesartan on attenuating ventricular remodeling and aldosterone production in the heart of rats. Irbesartan reduced the left ventricular mass index, collagen composition, and aldosterone levels, while enhancing cardiac function in myocardial infarction rats. In vitro, Ang II time-dependently stimulated aldosterone secretion and CYP11B2 mRNA expression in cardiac microvascular endothelial cells. Additionally, Ang II significantly upregulated p-CREB protein levels. Here are my comments related to this manuscript.
-In the abstract section, please describe the results with P values and indicate the strain of rats used in this study.
-The authors should describe the results section with P values. In addition, the scales are missing in the micrographs in Figure 1 (panel A). It seems that the percentage of collagen volume fraction is not representative according to micrographs (there is less fibrosis). Clarify whether micrographs are magnified to 100x. Why was the size of the infarct not assessed?
-The Figure 1 should have the expression of collagen I by western blot.
-Please assess myocardial hypertrophy by evaluating the cross-sectional area of ​​cardiomyocytes in the different groups.
-Clarify in legend of Figure 2 whether data are expressed as mean ± SD.
-For Figure 4, the authors should add a cell viability experiment according to different concentrations of Ang II.
-In Figure 6, blots for CREB without phosphorylation are missing.
-In materials and methods clarify whether sham groups underwent thoracotomy and pericardiotomy.
-Please add more detail on how the echocardiography was evaluated.
-Clarify whether cell treatments were in presence or absence of 10% fetal bovine serum. How many cells were per well-plate? Please add more detail related to these experiments. What was the vehicle for KN93?
-How was total RNA integrity evaluated?
Round 2
Reviewer 3 Report
Comments and Suggestions for Authors
In my opinion, the authors should perform the experiments suggested in revised version 1 of the manuscript. These experiments are important to give more clarity to the results.
Round 3
Reviewer 3 Report
Comments and Suggestions for Authors
I have no comments.
Author Response
Dear reviewer:
We should like to express our appreciation to you for suggesting how to improve our paper. We have studied your comments carefully in revised version 1 and 2 of the manuscript. Without doubt, we would benefit a lot from your precious comments, which could be helpful to our future studies.
Thank you!
